# Deriving Implications for Care Delivery in Parkinson’s Disease by Co-Diagnosing Caregivers as Invisible Patients

**DOI:** 10.3390/brainsci11121629

**Published:** 2021-12-10

**Authors:** Franziska Thieken, Marlena van Munster

**Affiliations:** 1Department of Neurology, University Hospital of Marburg, Baldingerstraße, 35043 Marburg, Germany; munster@med.uni-marburg.de; 2Faculty of Medicine, Philipps-Universität Marburg, Biegenstraße 10, 35037 Marburg, Germany

**Keywords:** Parkinson’s disease, informal caregiver, caregiving, caregiver burden, personalized care, co-diagnosis

## Abstract

For persons with Parkinson’s disease, the loss of autonomy in daily life leads to a high level of dependency on relatives’ support. Such dependency strongly correlates with high levels of perceived stress and psychosocial burden in informal caregivers. Global developments, such as demographic change and the associated thinning infrastructure in rural areas cause a continuously growing need for medical and nursing care. However, this need is not being adequately met. The resulting care gap is being made up by unpaid or underpaid work of informal caregivers. The double burden of care work and gainful employment creates enormous health-related impairments of the informal caregivers, so that they eventually become invisible patients themselves. Expectedly, those invisible patients do not receive the best care, leading to a decrease in quality of life and, in the end, to worse care for PD patients. Suggested solutions to relieve relatives, such as moving the person affected by Parkinson’s to a nursing home, often do not meet the wishes of patients and informal caregivers, nor does it appear as a structural solution in the light of demographic change against an economic background. Rather, it requires the development, implementation and evaluation of new, holistic approaches to care that make invisible patients visible.

## 1. Introduction

Although Parkinson’s disease (PD) is already one of the most common neurodegenerative disorders, the prevalence of PD is expected to double by the year 2040, and that is due to the aging of the susceptible group [1]. In PD, neural degeneration and loss of dopaminergic cells in the substantia nigra cause a lack of dopamine, which ultimately leads to impaired motor functioning (tremor, rigidity, bradykinesia, postural instability) and a variety of non-motor symptoms [2,3]. In the progression of the disease, motor and non-motor symptoms as well as the need for daily support increase, which in turn influences emotional well-being and social functioning [4]. Persons with PD (PwPDs) experience a loss of autonomy in daily life (e.g., in dressing, personal hygiene, nutrition, mobility, taking medication, etc.), which leads to a high level of dependency on informal caregivers’ support. Informal caregivers are defined as people providing any help to older family members, friends and people in their social network, living inside or outside of their household [5]. Therefore, informal caregivers include relatives as well as friends and neighbors. Looking at the global demographic developments, the structural organization of health systems, as well as the general requirements and challenges of living with PD, a few observations can be made that affect informal caregivers. Consequently, it may be necessary to visualize informal caregivers as invisible patients and included them more centrally in care planning in order to establish sustainable care models.

## 2. Observations

### 2.1. Informal Carers Take on Important Tasks in the Day-to-Day Care of Parkinson’s Patients and Need Support

Despite all the burdens, most informal caregivers strive to provide care in the home environment as long as possible [6]. For example, in 2019 home-based care was provided to eighty percent of the 4.1 million care recipients in the German healthcare system [7]. Seventy percent of the informal caregivers provided care without professional assistance, while only thirty percent employed an outpatient care service. Comparable numbers can also be found in other healthcare systems [8,9]. Thus, it can be observed that a high proportion of care is provided informally from the care recipient’s social network with little to no financial compensation [10].

Within the scientific literature four main reasons for care recipients’ and informal caregivers´ preference of home care are observable: First, a perceived lack of quality of nursing homes; second, the excessive costs of outpatient or inpatient care; third, to the care recipients wish to remain in their homes as long as possible; fourth, the care recipients preference to receive care from a familiar person [11,12]. Although most informal caregivers have no formal care qualifications, their care activity is well valued and highly appreciated by the PwPDs [13,14]. However, PD related changes often require challenging adaptation in activities of daily living, not only by the patient itself, but also by the informal caregiver. In addition to being relationship-oriented, care provision is communication-oriented and time-intensive, therefore it cannot be shortened or standardized without losing quality [15,16,17].

### 2.2. Taking on the Role of an Informal Caregiver Changes Self-Perception and Poses a Challenge to Physical and Mental Health

At the onset of chronic Parkinson’s disease, informal caregivers are usually burdened by the expected future effects of PD. Sharing of experiences with other relatives and relative-specific information from the treatment team has not yet been standardized [18]. In disease progression, motor and non-motor symptoms, as well as the need for daily support increase. Due to increasing need for supervision and emotional aspects of the caregiver relationship, non-motor symptoms (such as depression [19], anxiety [20], apathy [21], cognitive impairment [22], psychosis [23], impulse-control disorders [21], sleep [24] and pain [25]) make a greater contribution to caregiver burden than motor symptoms. The patients’ loss of autonomy in daily life leads to a high level of dependency on informal caregivers’ support, which strongly correlates with the perceived stress of informal caregivers [26,27]. Increasing perceived stress and role change gradually leads to a change in the demands of support needs. Thus, informal caregivers among other things need support in stress management, coping with emotional distress, information on receiving social support and education programs [18,28].

Informal caregivers often perceive the change in their role from mere friends or relatives to caregivers as a change in their social position—a change due to which potential conflicts between PwPDs and the caregiver family may ensue [29]. Due to the moral responsibility for the care recipient and the subordination of their own life to the flexibility requirements of the care situation, informal caregivers often live under unsecured living conditions and have no access to privately funded support measures [28]. Liberation of care situation is often hardly possible, because a reduction of the burden could mean a social decline or a lack of old-age security [30]. Relatives thus quickly become “prisoners of love” [31], sacrificially disregarding necessary self-care. Without a reduction in daily stress levels through self-care, prolonged symptoms of stress can develop into serious exhaustion. For example, thirty-eight percent of primary informal caregivers reported fatigue with no hope of recovery. Thirty percent felt trapped in the role as caregiver, for twenty percent caregiving was often too stressful, for twenty-three percent caregiving negatively affected friendship relationships, and nineteen percent had fears about the future and their livelihood [11,12]. Among informal caregivers, exhaustion syndromes are widely spread and reflected in a wide range of mental disorders like depression, adjustment disorders, and addictive disorders [9]. However, a large workload can affect the physical and mental health of informal caregivers in addition to sleep quality with serious effects on the quality of life [32,33]. A decreased quality of life in informal caregivers may negatively impact patient care itself [28,34]. For example, Schulz et al. showed that emotional stress in informal caregivers is an independent risk factor for patient’s mortality [17].

### 2.3. Informal Caregivers Require Legal Support Framework

From the view of a profit-oriented society, caring activities are not profitable and represent an enormous cost factor. In health and economic policy terms, informal caregivers of PwPDs have been described as system breakers, which describes people who overtax the system. However, it may be argued that the caregivers do not become system breakers, but the system turns them into system breakers. High work intensity, little room to maneuver, low acknowledgment of care activity and a lack of recreation time combined with high demands and a lack of social support increase the risk of psychological impairment and disorders in informal caregivers [35,36,37].

Thus, there is a need for structural solutions. In some countries, such as Germany, France or England, the informal caregivers’ care activity is counted toward the pension in national insurance credits and, depending on the type and severity of the respective impairment, remunerated with a non-cost-covering and lump-sum Carer’s Allowance [8,38,39]. Countries like the USA do not have regular financial compensation, but instead there are regional differences in US state’s supplemental programs [9]. Additionally, several healthcare systems offer paid leave for informal caregivers [9,38]. However, these measures do not meet the needs of informal caregivers of patients with chronic disease progression such as PD. PwPDs require continuous care throughout their lifespan which cannot be covered by paid leave. Therefore, informal caregivers of patients with PD often cannot be available to the labor market with the desired flexibility. Due to the pressure to perform, many informal caregivers of working age finally reduce working hour or withdraw from the labor market [16,40,41]. Therefore, the care of chronically ill persons often represents a heavy financial burden for informal caregivers [28]. As a result of the reduction or the complete loss of work, other family members may be threatened by a collapse of the living arrangement and experience stigmatization within society.

### 2.4. The Challenge of Being Informal Carer in Times of Crisis

The care deficit became particularly visible during the COVID-19 pandemic. The onset of the pandemic in spring 2020 significantly increased the demands on informal caregivers since formal care provisions were partly lost and had to be absorbed by the informal caregivers [34]. A lack of opportunities to provide sufficient care activity combined with an insufficient supply of affordable care services led to forty percent of informal caregivers reporting an additional burden and thirty-one percent felt overwhelmed [42]. Without such volunteerism care recipients, who were most helplessly exposed to the pandemic, would have been hit even harder. However, initial studies also show that this effort has left its mark on those affected and that existing challenges, such as a strong mental stress on informal caregivers, have intensified [34,43].

## 3. Conclusions and Relevance

Although several observations can be made that the diagnosis of PD deeply affects informal caregivers as well, they seem to be left alone in their despair. Due to the multi-dimensional character of their burden, informal caregivers often become invisible patients [28], who, more often than not, receive no attention during medical consultations. Since the respective setting is not able to offer the appropriate help, they are passed on.

Care provision for PD dares a fundamental change of perspective by placing the human needs of those affected, in all their diversity, in the center of consideration. From the observations we have presented, it becomes visible that is important to look at the individual caregiver at onset of PD, to recognize their need for support and, at best, to preventively guide them into a needs-oriented support system. Thus, framework conditions should be created that enable people in need of support to gain influence on how they are treated or cared for and enable relatives to care for themselves and others.

Such framework conditions may also be a prerequisite for establishing resilient healthcare systems. A growing need for PD care, as well as the potential occurrence of unforeseeable crises makes it clear that resources must be better used in the future. Failure to address the needs of informal caregivers poses a long-term social problem. A society is measured by its ability to guarantee sustainable and save conditions. Therefore, ideas and concepts must be developed that show a perspective for the future and at the same time already include steps for change today.

Thus far, a few lighthouse supporting programs have provided hope for improving the status quo [44,45]. For example, a pilot project in Sweden was able to demonstrate that a reduction in working hours of informal caregivers can increase motivation and satisfaction and at the same time improve the quality of care [46]. However, it is necessary that these diverse ideas, which arise as a reaction to grievances and social suffering in everyday life, gain strength by becoming more visible. People in permanently insecure and stressful living conditions often have hardly any strength left to engage and to organize themselves. Those resignation and withdrawal could be replaced by courage, energy and self-confidence trough joint action.

The first step may be a co-diagnosis for informal caregivers, so that the previously invisible patients of chronic PD can gain visibility. Co-diagnosis should be a newly created diagnosis for informal caregivers, assigned by treating physicians to express the extra-ordinary burden and associated disorders of cares of chronic diseases. With the introduction of this co-diagnosis, informal caregivers may receive the attention of healthcare providers and a needs-oriented supporting programs for the dyad of PwPD and informal caregiver patients could emerge. For example, care provisions for informal caregivers could include regular medical visit, supportive advice regarding the challenges of everyday life with PD and the discussion of their perceived burden with the treatment team. In addition, caregivers can receive needs-based access to psychological support, physical therapy services and rehabilitation to offset health risk. Additionally, a co-diagnosis may help informal caregivers to gain the long overdue acceptance in society and become more visible on the political agenda. For example, shortening the years of pension contributions for caregivers and higher regulated tax benefits could become central elements in the political debate to recognize the permanent caregiving activity for the chronically ill family member.

Thus, it may be concluded that making informal caregivers visible through co-diagnosis may be a first step towards a more sustainable and holistic care for PD.

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
