# Peer review of "Deriving Implications for Care Delivery in Parkinson’s Disease by Co-Diagnosing Caregivers as Invisible Patients"

_brainsci, 2021, doi:10.3390/brainsci11121629_

Round 1
Reviewer 1 Report
The authors reviews and discuss the issues about informal caregivers (= invisible patients) of Parkinson's disease patients. They cites and reviews previous studies from a variety of perspectives. The theme is a major social issue and a topic that could be of interest to readers.
The issues of the current manuscript is influenced by many social factors. While citing several previous studies, it may be difficult to generalize all issues to the world outside of the previous studies or reports. However, the authors have taken this into consideration and have written in a conservative manner. Overall, it does not seem to be an overstatement.
Is "..." in lines 35 and 37 necessary? Simply, "Informal caregivers are defined as people providing any help to older family members, friends and people in their social network, Living inside or outside of their household" would not be misleading.
Author Response
Dear Reviewer 1,
we have now revised the manuscript entitled, " Deriving implications for care delivery in Parkinson´s disease by co-diagnosing caregivers as invisible patients." according to the received comments of three expert referees. We are writing to you to resubmit our manuscript for consideration in the Brain Sciences. We found the reviewers’ comments to be helpful in revising the manuscript and have carefully considered and responded to each suggestion.
We address each comment the reviewer made below, numbering the reviewers’ comments and prefacing our responses by "Author's Response". Corresponding changes are highlighted in the manuscript text in the revised and tracked file.
Reviewer's Comment 1: Is "..." in lines 35 and 37 necessary? Simply, "Informal caregivers are defined as people providing any help to older family members, friends and people in their social network, Living inside or outside of their household" would not be misleading.
Author’s Response 1 (Page 1): In response to the reviewer’s comments, we have deleted the quotes in our revised version.
We very much hope the revised manuscript is now suitable for publication in Brain Sciences.

Reviewer 2 Report
Thank you for the opportunity to review this manuscript. Please clarify the idea of utilizing "a co-diagnosis for informal caregivers." (lines 154-155). The following questions could help clarify the proposed idea: a) what kinds of diagnoses could be given to informal caregivers? and b) which providers will diagnose those caregivers?
Author Response
Dear Reviewer 2,
we have now revised the manuscript entitled, " Deriving implications for care delivery in Parkinson´s disease by co-diagnosing caregivers as invisible patients." according to the received comments of three expert referees. We are writing to you to resubmit our manuscript for consideration in the Brain Sciences. We found the reviewers’ comments to be helpful in revising the manuscript and have carefully considered and responded to each suggestion.
We address each comment the reviewer made below, numbering the reviewers’ comments and prefacing our responses by "Author's Response". Corresponding changes are highlighted in the manuscript text in the revised and tracked file.
Reviewer's Comment 1: Please clarify the idea of utilizing "a co-diagnosis for informal caregivers." (lines 154-155). The following questions could help clarify the proposed idea: a) what kinds of diagnoses could be given to informal caregivers? and b) which providers will diagnose those caregivers?
Authors's Response 1: In response to the reviewer’s comments, we clarified this proposal by adding the following text passages on page 4:
“Co-diagnosis should be a newly created diagnosis for informal caregivers, assigned by treating physicians to express the extra-ordinary burden and associated disorders of cares of chronic diseases.”
We very much hope the revised manuscript is now suitable for publication in Brain Sciences.

Reviewer 3 Report
I have been reading carefully this opinion piece entitled "Deriving implications for care delivery in Parkinson's disease by co-diagnosing caregivers as invisible patients"
And it seemed to me that it is very well structured and raises very important problems regarding the health of the caregivers and the quality of life if they want to continue caring, for which they need more support from public aid.
I recommend that the authors include some aspects related to the phase of the disease and the burden on the caregiver (include some reference to these studies). It is also important to address the subject of the patient's pain and how it can influence the task of care.
It is a non-motor symptom that can promote the onset of depression in Parkinson's patients and can also make care tasks difficult for the informal caregiver.
You can mention it in part 2.2 Taking on the role of an informal caregiver changes self-perception and poses a challenge to 66 physical and mental health
I recommend studies in this regard that you can read and cite as a basis:
- Santos-García, D., Abella-Corral, J., Aneiros-Díaz, Á., Santos-Canelles, H., Llaneza-González, M. A., & Macías-Arribi, M. (2011). Pain in Parkinson's disease: prevalence, characteristics, associated factors, and relation with other non motor symptoms, quality of life, autonomy, and caregiver burden. Revista de Neurologia, 52(7), 385-393.
It is also important to highlight at this point the insomnia that informal caregivers can suffer:
Corey, K. L., McCurry, M. K., Sethares, K. A., Bourbonniere, M., Hirschman, K. B., & Meghani, S. H. (2020). Predictors of psychological distress and sleep quality in former family caregivers of people with dementia. Aging & mental health, 24(2), 233-241.
Simón, M. A., Bueno, A. M., Otero, P., Blanco, V., & Vázquez, F. L. (2019). Caregiver burden and sleep quality in dependent people’s family caregivers. Journal of clinical medicine, 8(7), 1072.
I encourage the authors to add regarding to financial aid, it would be good to propose a reduction in years of contributions for caregivers to obtain the maximum retirement contribution and higher regulated tax benefits in all countries of the European Union. You can check if there is any attempt by any committee of the European Parliament. Other health benefits would be those that can offset the health risk, such as free access to spas, physiotherapy services, etc. regulated from the national health systems of each member state of the European Union.
Some references are incomplete like:
- Hurh, K., et al., The Impact of Transitions in Caregiving Status on Depressive Symptoms among Older Family Caregivers: Findings 222 from the Korean Longitudinal Study of Aging. Int J Environ Res Public Health, 2020. 18(1). 

Or they are cut off in the text and should be followed:
- Bianchi, S.M., N. Folbre, and D.A. Wolf, Unpaid care work. In
(Ed.), For love and money: Care provision in the United States
York, NY: Russell Sage Foundation.
Some errors when leaving more space between words:
- Fasang, A., S. Aisenbrey, and K. Schömann, Fasang, A., Aisenbrey, S., & Schömann, K. (2013). Women’s retirement income in 214 Women’s retirement income in Germany and Britain. 2013, European Sociological Review. 

Links without hyperlink in the whole link:
- DIE PFLEGE ÄLTERER PERSONEN IN DEUTSCHLAND, FRANKREICH UND DER SCHWEIZ. 2018; Available 225 from: https://www.trisan.org/fileadmin/PDFs_Dokumente/2018-05-Themenheft_Pflege-älterer-Personen_DE.pdf. 

Author Response
Dear Reviewer 3,
we have now revised the manuscript entitled, "Deriving implications for care delivery in Parkinson´s disease by co-diagnosing caregivers as invisible patients." according to the received comments of three expert referees. We are writing to you to resubmit our manuscript for consideration in the Brain Sciences. We found the reviewers’ comments to be helpful in revising the manuscript and have carefully considered and responded to each suggestion.
We address each comment the reviewer made below, numbering the reviewers’ comments and prefacing our responses by "Author's Response". Corresponding changes are highlighted in the manuscript text in the revised and tracked file.
Reviewer's Comment 1: I recommend that the authors include some aspects related to the phase of the disease and the burden on the caregiver (include some reference to these studies). It is also important to address the subject of the patient's pain and how it can influence the task of care.
It is a non-motor symptom that can promote the onset of depression in Parkinson's patients and can also make care tasks difficult for the informal caregiver.
You can mention it in part 2.2 Taking on the role of an informal caregiver changes self-perception and poses a challenge to physical and mental health
I recommend studies in this regard that you can read and cite as a basis:
- Santos-García, D., Abella-Corral, J., Aneiros-Díaz, Á., Santos-Canelles, H., Llaneza-González, M. A., & Macías-Arribi, M. (2011). Pain in Parkinson's disease: prevalence, characteristics, associated factors, and relation with other non motor symptoms, quality of life, autonomy, and caregiver burden. Revista de Neurologia, 52(7), 385-393.
It is also important to highlight at this point the insomnia that informal caregivers can suffer:
Corey, K. L., McCurry, M. K., Sethares, K. A., Bourbonniere, M., Hirschman, K. B., & Meghani, S. H. (2020). Predictors of psychological distress and sleep quality in former family caregivers of people with dementia. Aging & mental health, 24(2), 233-241.
Simón, M. A., Bueno, A. M., Otero, P., Blanco, V., & Vázquez, F. L. (2019). Caregiver burden and sleep quality in dependent people’s family caregivers. Journal of clinical medicine, 8(7), 1072.
Author’s Response 1 (page 2 and 3): In response to the reviewer’s comments, we refined especially part 2.2 by describing the needs in different phases of disease. In addition, we described the importance of non-motor symptoms (e.g. pain and sleep) on the perception of stress.
Reviewer's Comment 2: I encourage the authors to add regarding to financial aid, it would be good to propose a reduction in years of contributions for caregivers to obtain the maximum retirement contribution and higher regulated tax benefits in all countries of the European Union. You can check if there is any attempt by any committee of the European Parliament. Other health benefits would be those that can offset the health risk, such as free access to spas, physiotherapy services, etc. regulated from the national health systems of each member state of the European Union.
Author’s Response 2 (Page 4 f.): Thank you for your comment. We have taken your feedback as an opportunity to expand the requirements in our last section in accordance with your feedback.
Reviewer's Comment 3: Some references are incomplete like:
- Hurh, K., et al., The Impact of Transitions in Caregiving Status on Depressive Symptoms among Older Family Caregivers: Findings 222 from the Korean Longitudinal Study of Aging. Int J Environ Res Public Health, 2020. 18(1). 

Or they are cut off in the text and should be followed:
- Bianchi, S.M., N. Folbre, and D.A. Wolf, Unpaid care work. In
(Ed.), For love and money: Care provision in the United States
York, NY: Russell Sage Foundation.
Some errors when leaving more space between words:
- Fasang, A., S. Aisenbrey, and K. Schömann, Fasang, A., Aisenbrey, S., & Schömann, K. (2013). Women’s retirement income in 214 Women’s retirement income in Germany and Britain. 2013, European Sociological Review. 

Links without hyperlink in the whole link:
- DIE PFLEGE ÄLTERER PERSONEN IN DEUTSCHLAND, FRANKREICH UND DER SCHWEIZ. 2018; Available 225 from: https://www.trisan.org/fileadmin/PDFs_Dokumente/2018-05-Themenheft_Pflege-älterer-Personen_DE.pdf. 

Author’s Response 3: Many thanks for the hint. The references have been revised again.
We very much hope the revised manuscript is now suitable for publication in Brain Sciences.

Round 2
Reviewer 2 Report
Thank you for addressing my comments in the revised manuscript.